# Cattle Zoonotic and Non-Zoonotic Tick-Borne Pathogens in Europe—A Retrospective Analysis of the Past 15 Years

**DOI:** 10.3390/ani15101408

**Published:** 2025-05-13

**Authors:** Diana Hoffman, Ioan Cristian Dreghiciu, Ion Oprescu, Mirela Imre, Tiana Florea, Anamaria Plesko, Sorin Morariu, Marius Stelian Ilie

**Affiliations:** Department of Parasitology and Parasitic Disease, Faculty of Veterinary Medicine, University of Life Sciences “King Mihai I” from Timisoara, 119, Calea Aradului, 300645 Timisoara, Romania; ioan.oprescu@fmvt.ro (I.O.); mirela.imre@usvt.ro (M.I.); tijana.florea@usvt.ro (T.F.); plesko.anamaria.fa@usvt.ro (A.P.); sorin.morariu@fmvt.ro (S.M.); mariusilie@usvt.ro (M.S.I.)

**Keywords:** cattle, tick-borne diseases, *Anaplasma*, *Borrelia*, *Babesia*, *Theileria*, Europe

## Abstract

The study highlights the growing challenge posed by vector-borne diseases, especially ticks, in cattle populations across Europe. The primarily identified vectors are *Ixodes ricinus* and *Haemaphysalis punctata*, which spread pathogens such as *Anaplasma* spp., *Babesia* spp., *Theileria* spp., and *Borrelia burgdorferi*. The occurrence of these infections is on the rise, influenced by factors like environmental shifts and the movement of livestock. Diagnostic methods such as polymerase chain reaction, enzyme-linked immunosorbent assay, and genetic sequence analysis are commonly used to detect these diseases. The findings underscore the need for improved monitoring protocols on farms and in areas with increased vector-related activity, promoting a unified “One Health” strategy to address the interconnected risks regarding animal, human, and environmental health.

## 1. Introduction

Vector-borne diseases (VBDs) are widely distributed across the globe. The majority of these diseases, classified as metazoonoses, can be transmitted to both humans and domestic or wild animals through invertebrate arthropod hosts, particularly ticks. According to the World Health Organization (WHO) (2 March 2020), vector-borne diseases account for more than 17% of all infectious diseases worldwide, resulting in over 700,000 human fatalities annually [1,2].

The global “One Health” concept brings together three constantly relevant and inherently interconnected components: human health, animal health, and the environment. It emphasizes the identification of risk factors, modes of transmission, as well as the development of prevention and control measures for both zoonotic and non-zoonotic diseases. This triad can be disrupted by factors such as population growth, globalization, urbanization, climate change, travel, and changes in land use, all of which contribute to the emergence of zoonotic diseases, vector-borne diseases, and food- or waterborne illnesses. Nevertheless, this integrated system can be strengthened through the involvement of wildlife specialists, veterinary and human medical professionals, practitioners of alternative medicine, public and environmental health experts, dentists, physicists, biomedical engineers, biochemists, and other specialists [3,4,5].

From an economic perspective, the impact of tick-borne diseases extends to four major populations: humans, economically significant livestock, companion animals, and wildlife. The annual economic losses associated with these diseases amount to billions of dollars. Despite the wide range of climatic conditions that support tick abundance and the development of all life cycle stages, significant efforts have been directed toward reducing parasitism with these arthropods. In recent decades, multiple chemical control strategies have been implemented, particularly in developing countries, to mitigate the effects of tick infestations [6,7,8]. However, the methods for controlling both ticks and the diseases they transmit remain unclear [6,7,8].

Anaplasmosis was first recognized in 1990, and nine years later, it was officially included on the list of notifiable diseases. The disease is widely distributed in tropical and subtropical regions worldwide. In cattle, infection typically occurs through tick bites, although transmission can also occur via biting flies and contaminated veterinary instruments, such as needles, dehorning tools, castration equipment, and ear-tagging pliers. Additionally, transplacental transmission has been documented in *Anaplasma marginale*. Evidence suggests that cattle can undergo frequent reinfections, indicating a potential reservoir role for these animals. Bovine anaplasmosis is an infectious disease caused by *Anaplasma marginale*, *Anaplasma phagocytophilum* (a zoonotic species), *Anaplasma centrale*, and *Anaplasma bovis*. In cattle, the disease may manifest as asymptomatic or present with progressive hemolytic anemia, accompanied by symptoms such as fever, jaundice, anorexia, hypersalivation, diarrhea, dyspnea, frequent urination, and weight loss. In pregnant animals, abortion may also occur [9].

Babesiosis is a parasitic disease frequently reported in tropical and subtropical regions, ranking second in prevalence after trypanosomiasis [10]. The etiology of bovine babesiosis is complex, with several species responsible for the disease worldwide, including *Babesia bigemina*, *Babesia bovis*, *Babesia divergens*, *Babesia major*, *Babesia occultans*, and *Babesia argentina*. Among these, *Babesia bigemina* (African red water) is the most widespread, while *Babesia bovis* (Asiatic red water) is also of significant concern [11,12]. The clinical presentation is characterized by severe hemolytic anemia, fever or low-grade fever, pale or icteric mucous membranes, anorexia, hypogalactia/agactia, impaired rumination, forestomach atony, tachypnea, tachycardia, and hemoglobinuria. Immunosuppressed cattle or those deprived of grazing access during their first year of life exhibit a higher susceptibility to infection. The morbidity and mortality rates vary significantly and are influenced by multiple factors, including recent treatments applied in the region, previous exposure to the parasite, cattle age and breed, vaccination status, geographic location, sex, herd size, seasonal management practices, insect abundance, livestock density, presence of other domestic animals on the farm, tick infestation levels in both cattle and their shelters, and pasture management strategies [10].

Theileriosis is a parasitic disease affecting the blood and lymphatic system, transmitted by ticks of the *Ixodidae* family, specifically from the *Amblyomma*, *Haemaphysalis*, *Hyalomma*, and *Rhipicephalus* genera. *Theileria* species that infect cattle include *Theileria annulata*, *Theileria parva*, *Theileria mutans*, the *Theileria orientalis* complex (*orientalis*/*sergenti*/*buffeli*), *Theileria tarurotragi*, *Theileria velifera*, *Theileria sinensis*, and *Theileria* sp. Yokoyama, a newly discovered species closely related to *T. annulata*. Among these, the most frequently reported species in Europe include *T. annulata*, *T. orientalis*, *T. parva*, *T. buffeli*, *T. sergenti*, and *T. sinensis*. *T. orientalis*, *T. buffeli*, and *T. sinensis* are considered responsible for inducing mild or asymptomatic forms of the disease. Although theileriosis is commonly found in domesticated cattle in tropical and subtropical countries, its clinical presentation can sometimes be atypical. However, in symptomatic cases, affected animals may exhibit intravascular hemolysis, tachypnea, tachycardia, dyspnea, dark-colored urine, and hyperbilirubinemia. Lymphadenopathy may also be present, along with leukopenia, neutropenia, and either hypoproteinemia or hyperproteinemia [13].

In Europe, *Borrelia burgdorferi* sensu lato is transmitted by *Ixodes* ticks. The *Borrelia afzelii* species has rarely been reported in cattle, where it induces symptoms such as erythema, fever, and lameness [14,15].

However, three *Borrelia* species are associated with distinct disorders in cattle: *Borrelia burgdorferi*—Lyme disease; *Borrelia theileri*—Bovine tick-borne spirochetosis; and *Borrelia coriaceae*—Epizootic bovine abortion. Experimental infections in cattle with *B. burgdorferi* sensu stricto have been conducted, whereas *Borrelia garinii* and *B. afzelii* did not induce clinical signs [1,15]. All of these are spirochetes belonging to the *Spirochaetaceae* family, *Spirochaetales* order, and *Borrelia* genus. The name of the first mentioned species (*B. burgdorferi*) originates from American microbiologist Willy Burgdorfer, who identified the species in deer ticks, *Ixodes dammini* [14,15].

In cattle, borreliosis presents with dermatitis, interdigital erythema, fever, lameness, polyarthritis, meningoencephalitis, weight loss, mammary erythema, and reduced milk production [15].

Thus, within the framework of the One Health concept, among the pathogens transmitted by ticks, only part of them are zoonotic, while the remainder are non-zoonotic. For instance, in the case of babesiosis, only three species are known to be transmissible from cattle to humans: *Babesia microti, Babesia divergens,* and *Babesia venatorum* (also known as *Babesia* sp. EU1) [16,17,18,19].

Regarding theileriosis, no zoonotic potential has been reported to date; however, the disease has a substantial impact on the livestock industry, particularly in tropical and subtropical regions [18].

Two species—*Anaplasma phagocytophilum* and *Anaplasma capra*—are considered to be zoonotic and are thus responsible for causing human anaplasmosis. *A. phagocytophilum* was first identified in sheep in 1932, and it was not until 62 years later that it was confirmed in humans. *A. capra* was initially detected in sheep and cattle in 2017 and 2018, respectively, and later in cattle from Angola in 2021. Its zoonotic potential has been confirmed in a single study conducted in China by Li et al., 2015 [20,21].

Human borreliosis, or Lyme disease, was first described in 1977 by Dr. Alan Steere and colleagues as an infectious disease. The zoonotic agents responsible for the disease are bacteria such as *Borrelia burgdorferi*, which is also found in cattle, and *Borrelia mayonii*, which has not been identified in large ruminant populations. *B. burgdorferi* is commonly associated with Lyme disease in the United States, whereas *B. mayonii* is less prevalent, with higher rates of occurrence in the northeastern, mid-Atlantic, and upper Midwestern regions of North America [22,23].

Therefore, this study provides an overview of the current distribution of tick-borne diseases in cattle, along with a review of the most frequently implicated tick species in disease transmission and the zoonotic pathogens reported in Europe over the past 15 years. To implement effective measures aimed at reducing or even preventing the future spread of these diseases, systematic surveillance will be essential. This includes the collection, analysis, and dissemination of information regarding the most significant infections affecting both human and animal health. This study focuses on four major tick-borne diseases commonly reported across Europe. These, listed in the order of frequency cited in the reviewed literature, are: anaplasmosis, babesiosis, theileriosis, and borreliosis.

## 2. Materials and Methods

### Literature Survey

For the development of this review, an extensive and comprehensive examination was conducted across three databases: PubMed Central^®^ (PMC), Science Direct, and Web of Science (WOS). The obtained results contributed to forming an overall perspective regarding the aspects outlined in the first section of this paper, namely tick-borne diseases in cattle, prevalence, and the most frequently reported parasite and bacterial species. To ensure consistency, the same keywords—“tick-borne disease in cattle in Europe”—were used across all three search engines. All selected articles were restricted to studies conducted exclusively in Europe and published between 2009 and 2024 (a 15-year timeframe). The selection process involved a quantitative analysis based on abstracts, followed by a qualitative analysis through an in-depth review of the relevant studies. All selected articles had to meet the following criteria (Figure 1) [1,24]:Timeframe: 2009–2024;Identification of tick species involved in disease transmission and the dissemination of etiological agents across Europe;Diagnostic methods employed;Pathogenic species frequently reported in cattle;Pathogenic species commonly identified in invertebrate hosts.

**Figure 1 animals-15-01408-f001:**
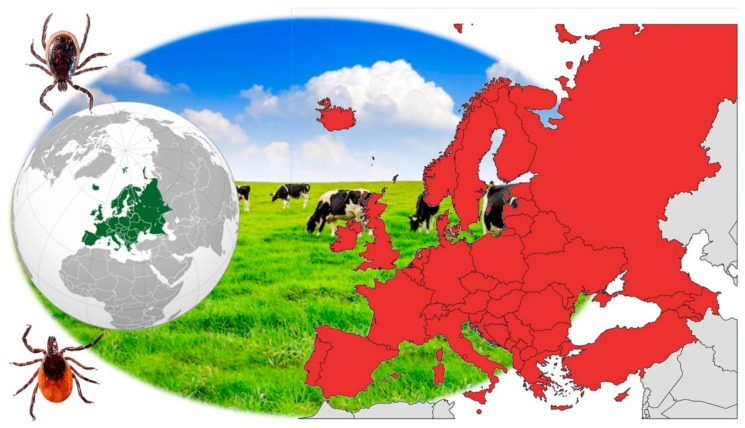
Key elements of the study. The global map highlights Europe (marked in green), the geographical area of interest in the study, which is shown magnified in a red box. The main components of pathogen transmission are illustrated: the involved vectors (ticks) and the target host species (cattle). Adapted from references [25,26,27].

## 3. Results

The search yielded the following results: on PubMed Central^®^ (PMC), a total of 108 articles were identified, of which only 19 met the inclusion criteria. On Science Direct, 2292 articles were found, but only 8 were deemed eligible. On Web of Science (WOS), from a total of 93 articles, only 9 fulfilled the required criteria. Overall, a final selection of 36 articles was obtained. Some of these studies contained information on multiple pathogenic agents.

Articles were excluded if they did not pertain to the targeted geographical region, lacked sufficient data, did not have full-text availability, were duplicate studies, or focused on other diseases transmitted by the relevant invertebrate hosts (Figure 2) [1].

### 3.1. Diagnostic Methods

A definitive diagnosis is established by correlating epidemiological, clinical, and pathological data with laboratory findings. Across the 36 articles reviewed, several diagnostic methods were reported as being used to confirm the presence of the pathogens. Laboratory techniques, ranked in descending order of frequency, included: PCR (polymerase chain reaction); real-time PCR; clinical signs-based diagnosis; multiplex real-time PCR; nested PCR; serological tests, such as ELISA, iELISA, and IFAT; as well as Giemsa-stained spleen smears and cloning. Additional valuable tools that contributed to a more accurate diagnosis of the diseases under investigation included surveys conducted among veterinarians and farmers, along with post-mortem examinations of carcasses [1,7,9,13,28,29,30,31,32,33,34,35,36,37,38,39,40,41,42,43,44,45,46,47,48,49,50,51,52,53,54,55,56,57,58,59,60].

Polymerase chain reaction (PCR) is a laboratory technique used to rapidly generate a large number of copies of a specific DNA fragment, producing millions or even billions of copies for detailed analysis. The PCR process involves the use of short synthetic DNA sequences, known as “primers”, which target a specific genomic segment for amplification. This is followed by multiple cycles of DNA synthesis, allowing for the exponential multiplication of the selected DNA fragment [61]. Polymerase chain reaction (PCR) is widely utilized in molecular diagnostics for identifying tick-borne pathogens. Conventional PCR demonstrates a sensitivity capable of detecting as low as six cells of *Borrelia burgdorferi* per reaction, achieving approximately 94% specificity in identifying infected ticks [62,63,64].

Real-time polymerase chain reaction/quantitative PCR (real-time PCR/qPCR) is frequently used to quantify gene expression by monitoring DNA amplification in real time throughout the reaction process. This technique enables the precise and sensitive detection of nucleic acids, making it a valuable tool in molecular biology, diagnostics, and research applications [65]. Real-time polymerase chain reaction (qPCR) enhances pathogen detection by allowing real-time quantification and amplification monitoring. Studies show that qPCR provides a sensitivity of up to 99.53%, coupled with exceptional specificity, effectively minimizing cross-reactions with non-target pathogens [66,67].

Multiplex quantitative PCR (Multiplex qPCR) is an optimal technique for various applications, including gene expression analysis, single-nucleotide polymorphism (SNP) genotyping, copy number variation (CNV) detection, identification of genetically modified organisms (GMOs), pathogen detection, and monitoring the efficacy of pharmacological treatments. This method allows for the simultaneous amplification and quantification of multiple target sequences in a single reaction, enhancing efficiency and reducing sample processing time [68]. Multiplex quantitative PCR (Multiplex qPCR) facilitates the simultaneous detection of multiple pathogens, streamlining diagnostics. A study conducted by Hojgaard et al. (2014) demonstrated that multiplex qPCR assays detecting *Borrelia burgdorferi*, *Anaplasma phagocytophilum*, and *Babesia microti* offer comparable efficiency and sensitivity to single-pathogen qPCR assays [69].

Nested polymerase chain reaction (Nested PCR) is employed when an enhanced sensitivity and/or specificity of PCR is required. This technique involves two successive rounds of amplification using two sets of primers, where the second set targets an internal region of the first amplicon. This approach significantly reduces non-specific amplification and improves the detection of low-abundance DNA targets Cold Spring Harbor protocols [70]. Nested polymerase chain reaction (Nested PCR) significantly improves sensitivity and specificity due to its two successive amplification rounds with distinct primer sets. Nested PCR has been shown to enhance the detection rate of *Borrelia burgdorferi* in tick samples, surpassing conventional PCR methods [71].

Nested quantitative PCR (Nested qPCR) combines the high sensitivity of nested PCR with the quantification capabilities of real-time PCR. Recent studies reported a sensitivity of 98.6% and specificity of 100% for detecting *Babesia duncani* in human and hamster blood samples, underscoring its diagnostic reliability [72,73].

ELISA (enzyme-linked immunosorbent assay) is a commonly used laboratory test for detecting antibodies in blood [74].

In a direct ELISA test, the primary detection antibody binds directly to the target protein. The advantages of this method include the elimination of cross-reactivity caused by secondary antibodies and a faster procedure due to the reduced number of steps compared to indirect ELISA. However, this technique also has disadvantages, such as lower sensitivity compared to other ELISA variants and higher reaction costs [75].

Indirect ELISA is a two-step ELISA method that involves two distinct binding processes: the first with the primary antibody and the second with the enzyme-labeled secondary antibody. In this technique, the primary antibody is incubated with the antigen, followed by incubation with the secondary antibody, which enhances detection sensitivity and signal amplification [76].

Indirect ELISA stands out for its higher sensitivity compared to the direct format. It is also more cost-effective and highly versatile, as it allows the use of a wide range of primary antibodies. The main drawback of this method is the potential for cross-reactivity involving the secondary detection antibodies [77].

Indirect fluorescent antibody test (IFAT) is a diagnostic assay based on the binding principle between the antigen and specific antibodies present in the patient’s serum. This method allows for the detection of IgM antibodies, which are indicative of an acute infection. IFAT utilizes a fluorescent-labeled secondary antibody to enhance visibility under a fluorescence microscope, making it a sensitive and widely used technique in infectious disease diagnostics [78].

The sensitivity of the indirect immunofluorescence assay (IFAT) varies depending on the investigated pathogen, the type of sample used, and the specific protocol applied. In general, IFAT is recognized for its high sensitivity and specificity in the serological diagnosis of various infections. For example, in the diagnosis of *Strongyloides stercoralis* in humans, IFAT demonstrated a sensitivity of 97.4% and a specificity of 97.9% at a serum titer of ≥1:20. In cattle infected with *Babesia bovis*, IFAT showed a sensitivity of 94.2% and a specificity of 100% for antibody detection in serum, outperforming ELISA in this context [79,80].

In conclusion, achieving a definitive diagnosis of a disease requires a comprehensive approach that integrates theoretical knowledge (epidemiological and clinical) with practical application (laboratory methods) (Table 1).

Of the total articles examined, 15 used the PCR method, 4 applied the RT-PCR method, and the remaining studies employed techniques such as ELISA (1 study), iELISA (1 study), IFAT (1 study), and Giemsa staining (1 study).

### 3.2. Species of Ticks

Across Europe, according to the latest studies, the distribution of ticks and the spread of tick-borne diseases are on an upward trajectory. Ticks are the second most significant vectors, following mosquitoes, in the transmission of pathogenic agents to livestock. These hematophagous ectoparasites have an ecology that encompasses host interactions, distribution patterns, and the influence of biotic factors, all of which contribute to a comprehensive understanding of species with veterinary significance. Following the literature search across the three selected databases, a higher prevalence of certain tick species was observed, while others were mentioned less frequently. The most frequently reported tick species, in descending order of mention, within the 36 relevant articles, are: *Ixodes ricinus*, *Hyalomma marginatum*, *Dermacentor marginatus*, *Rhipicephalus bursa*, *Dermacentor reticulatus*, *Haemaphysalis punctata*, *Rhipicephalus sanguineus*, *Rhipicephalus* (*Boophilus*) *annulatus*, *Hyalomma scupense*, *Hyalomma lusitanicum*, *Rhipicephalus turanicus*, *Rhipicephalus pusillus*, *Haemaphysalis inermis*, *Haemaphysalis parva*, *Haemaphysalis sulcata*, and *Haemaphysalis concinna*.

According to the reviewed studies, *I. ricinus* (70%) is highly abundant in mountainous pasture areas as well as in lowland regions, where Europe’s temperate climate prevails. As the most significant vector, *I. ricinus* plays a major role in the transmission of multiple diseases and is followed in vectorial importance by *D. reticulatus* and *D. marginatus*.

It is well established that *D. marginatus* is adapted to warmer and drier climates in southern latitudes, whereas *D. reticulatus* thrives in a moderately humid climate in more northern latitudes [3,7,9,13,28,29,32,34,35,36,37,40,42,43,44,45,47,50,52,53,54,57,58].

Establishing a correlation between tick vector species and the parasitic or bacterial pathogens they can transmit yields the following results:✓*Borrelia burgdorferi* sensu lato (*B. afzelii*, *B. garinii*, *B. burgdorferi* s.s., *B. valaisiana*, *B. lusitaniae* ș.a.) is transmitted by *I. ricinus* [7,32,36,42,54] and *H. marginatum* [45].✓*Borrelia miyamotoi*, first isolated in 1995 in Japan from the species *Ixodes persulcatus*, was later detected for the first time in the tick species *Dermacentor reticulatus* [40].✓*Anaplasma* spp. was isolated from *R. turanicus* [43].✓*A. phagocytophilum* was reported as being present in the following tick species: *I. ricinus* [7,9,32,42], *Haemaphysalis punctata, Rhipicephalus pusillus, Dermacentor marginatus*, *Hyalomma marginatum,* and *Rhipicephalus* spp. [34,44].✓*Anaplasma bovis* is transmitted by *Haemaphysalis punctata* [44].✓*Anaplasma marginale* was present in *Rhipicephalus bursa, Rh. Turanicus,* and *Hyalomma excavatum* ticks [58,82].✓*B. divergens*, *B. venatorum*, *B. divergens/capreoli,* and a species related to *B. odocoilei*, called *B. odocoilei-like,* are transmitted by hard ticks such as *I. ricinus* [9,28,29,47].✓*Babesia venatorum* was identified in *I. Ricinus* ticks [42].✓For *Babesia* spp., *R. turanicus*/*D. marginatus,* and *Hyalomma lusitanicum,* tick species could be the most important vectors [43].✓*Babesia bigemina* was detected in *Rhipicephalus annulatus* and *Rhipicephalus bursa* tick species [50].✓*Babesia bovis* was detected in *Rhipicephalus turanicus, Rh. bursa*, *H.excavatum*, and *H. anatolicum* ticks [82].✓*B. occultans* was detected in *H*. *excavatum* [82].✓*Theileria* spp. are mainly transmitted by *R. turanicus*/*D. marginatus* and *Hyalomma lusitanicum* [43].✓*Theileria annulata* was detected in *R. bursa*, *R. turanicus,* and *H. Excavatum* ticks [58,82].✓*Th. sergenti/buffeli/orientalis* were confirmed in *Rhipicephalus annulatus* ticks [50].

Therefore, the zoonotic and non-zoonotic diseases mentioned in this study are transmitted by hard ticks (family *Ixodidae*) (Table 2).

### 3.3. Tick-Borne Pathogens

The list of emerging tick-borne zoonotic pathogens identified within our survey and which represent a significant population throughout Europe include *Anaplasma* spp., *Babesia* spp., *Theileria* spp., and *Borrelia* spp. [54].

Thus, among the pathogens to be discussed in the following subsections, three are considered zoonotic: *Babesia* spp., *Borrelia* spp., and *Anaplasma* spp., while, to date, *Theileria* spp. has not been shown to be transmissible to humans.

#### 3.3.1. *Anaplasma* spp.

This review, based on 14 articles, highlights the most frequently studied and reported *Anaplasma* species in Europe, namely *A. phagocytophilum*, *A. marginale*, *A. bovis*, and *A. centrale*. The studies not only focused on their identification in the blood and carcasses of cattle but also confirmed their presence within various tick species. As a result, the zoonotic etiological agent responsible for inducing granulocytic anaplasmosis in dogs, horses, and humans, as well as tick-borne fever in domestic ruminants, *A. phagocytophilum*, has been reported in the following tick species: *I. ricinus*, *H. punctata*, *Rh. pusillus*, *H. marginatum*, and *D. marginatus* [29,32,33,34,38,40,42,43,44,48,51,54,55,58,83,84].

For *A. marginale*, the causative agent of acute anaplasmosis, a bovine syndrome characterized by progressive hemolytic anemia associated with abortion, fever, decreased milk production, weight loss, and, in some cases, death, confirmation was established in its vector: *Rh. bursa*. Other Gram-negative bacteria within the *Anaplasmataceae* family, reported in ticks, include: *A. centrale*—an intraerythrocytic species closely related to *A. marginale* [83], but less pathogenic, causing only mild clinical signs in cattle, identified in *H. punctata* and *A. bovis*, along with *A. centrale*, confirmed in *H. punctata* ticks [29,32,33,34,38,40,42,43,44,48,51,54,55,58,83,84].

McFadzean et al. (2023) reported the presence of co-infections involving *A. phagocytophilum*, while Andersson et al. (2017) discussed that 18% of animals testing positive for *Babesia* spp., as confirmed by PCR, also exhibited a simultaneous co-infection with *Anaplasma* spp. [29,33].

A comprehensive qualitative and quantitative analysis conducted across the three selected search engines, which served as the foundation for this study, identified 18 studies reporting the presence of *Anaplasma* spp. in both biological samples collected from cattle and within the invertebrate tick host. Among these 18 scientific papers, only 13 specified the prevalence of the disease in cattle populations, which ranged from 0.6% to 100%. Additionally, nine studies reported the prevalence of the etiological agent identified in ticks, with a recorded prevalence of 0.5% [29,32,33,34,38,40,42,43,44,48,51,54,55,58,83,84,85,86].

The obligate intracellular rickettsial pathogen *Anaplasma phagocytophilum* exhibits significant differences between Europe and North America due to the circulation of various strains and ecotypes. The prevalence of *A. phagocytophilum* has been studied in countries such as Sweden, the Czech Republic, Norway, and Switzerland. The lowest prevalence rate was reported in the Czech Republic at 5.5%, while the highest was recorded in Norway, reaching up to 100% in a group of clinically affected animals. Switzerland reported the highest prevalence of this species in asymptomatic cattle, at 63.0%. Other reported rates ranged from 5.5% in the Czech Republic to 85.7% in symptomatic cattle from Sweden [54].

Looking at other geographic regions, such as England (southwest, northwest, and northeast), central Spain, and two Belgian provinces—Liège and Walloon Brabant—prevalence rates in cattle with clinical signs were as follows: 20%, 1.8% ± 1.2% [7,51], 34.2% [44], and between 55.6% and 71.4% [44].

However, a definitive comparison of prevalence rates in large ruminants is difficult to establish due to significant seasonal variation, as results may be influenced by the time of year when animal sampling is conducted. For example, in a Swiss study, the seroprevalence in two groups of cattle varied from 16% before the grazing season to 63% at the end of the season [54].

Moreover, a potential cross-reactivity in serological tests with other *Anaplasma* species must be considered, such as between *A. marginale* in cattle and *A. ovis* in sheep. In Europe, *A. marginale* is primarily found in the Mediterranean region, but it has also been reported in Switzerland, Austria, Hungary, and Turkey (2.5%). *Anaplasma ovis* has been detected in France, Slovakia, and Hungary [54,82].

In Russia, Bursakov et al. (2019) reported a prevalence of 67% for either *Theileria* spp. or *A. marginale*, and also identified co-infections of *Theileria* spp. and *A. marginale* with a prevalence of 19% [41]. In Belgium, the overall prevalence of *Anaplasma* spp. was 15.6%, with the highest rates recorded in the provinces of Liège and Walloon Brabant, ranging from 44.4% to 42.7% [44].

The prevalence of specific bacterial species has also been investigated by Palomar et al. (2015), who reported in Spain a prevalence of 5.6% for *A. phagocytophilum* in *Haemaphysalis punctata* ticks collected from cattle and 1.4% for *Anaplasma centrale* in ticks of the same species and host [48].

In Turkey, the reported prevalence values of *Babesia bovis*, *Babesia occultans*, and *Theileria annulata* were 7.9%, 0.7%, and 5.8%, respectively [82].

#### 3.3.2. *Babesia* spp.

Bovine babesiosis is widely distributed in tropical and subtropical regions of the world. The World Organization for Animal Health (WOAH) has classified babesiosis under “List B” diseases, recognizing it as a condition capable of causing significant economic disruption if it emerges within cattle populations. Globally, an estimated 1.3 billion domestic animals are at risk of exposure to this disease [87,88,89,90]. Cernăianu (1958) reported the presence of the following *Babesia* species in bovid species (ox, buffalo, zebu): *Piroplasma bigeminun*, *Babesiella bovis*, *Babesiella major*, *Babesiella* (sin. *Françaiella*) *berbera*, *Babesiella* (sin. *Françaiella*) *argentina*, *Babesiella* (sin. *Françaiella*) *colchica*, *Françaiella caucasica*, and *Françaiella tarandi rangiferis* [91].

In the section entitled *Babesia* spp., we found a total of 18 articles reporting the identification of one or more species belonging to this genus. The *Babesia* species cited in the scientific literature include: *B. divergens*, *B. microti*, *B. occultans*, *B. venatorum*, *B. bigemina*, and *B. bovis*. These species are recognized by international health authorities (WOAH, formerly OIE) as being among the primary causative agents of bovine babesiosis [28,29,30,31,33,39,42,43,44,47,50,51,52,54,55].

Only 13 studies report data on the prevalence of the disease in cattle, with values ranging between 0.0% and 100%. In contrast, only 6 out of the 18 studies provide information on the prevalence of *Babesia* species in ticks, with a recorded prevalence of 0.38% [28,29,30,31,33,39,42,43,44,47,50,51,52,54,55,92,93]. Patial et al. (2021) reported the presence of *Babesia* species, particularly *B. divergens*, in blood samples collected from cattle as well as in ticks [94]. The confirmation of this species in European cattle, according to McFadzean et al. (2023) [29], is linked to climate change and land use for cattle grazing. The McFadzean et al. (2023) [29] study further states that “bovine babesiosis is geographically widespread across England and Wales, meaning a significant proportion of the cattle population is at risk of infection, with potential zoonotic transmission to humans”. Additionally, the same study confirms the presence of *Babesia* spp., with its identification based on “established blood smear examination techniques and a species-specific PCR method for *B. divergens*”. Furthermore, the study also reports the co-existence of *Anaplasma phagocytophilum* infections in affected cattle [29].

In opposition, Adjadj et al. (2023) and Springer et al. (2020) reported the absence of *Babesia* spp. in vector ticks. However, Bonnet et al. (2014) highlighted the zoonotic potential of several *Babesia* species identified in ticks, including *B. divergens*, *Babesia* spp. EU1, *B. microti*, and *B. major* [44,54,95].

Mysterud et al. (2018) [32] reported that, at the European level, particularly in eastern Norway, babesiosis exhibits a higher prevalence compared to anaplasmosis, making Norway the only European country where this trend has been identified. In contrast, for anaplasmosis, the most frequently mentioned bacterial etiological agent is *A. phagocytophilum*, which is found in a lower abundance [32].

In Romania, Ioniță et al. (2013) documented the identification of *B. occultans* in two tick genera, namely *D. marginatus* and *H. marginatum* [96].

Similarly, in Poland, Staniec et al. (2018) confirmed the presence of *B. occultans* EU376017 using real-time PCR diagnostic techniques [31].

In Portugal, Gomes et al. (2013) reported the presence of *B. bigemina*; however, the prevalence was very low [97].

Although babesiosis and anaplasmosis do not share the same etiological origin, both diseases can coexist within the same host, suggesting the presence of oxidative stress. This condition is generally defined as a disruption of the balance between oxidants and antioxidants. The imbalance may result either from an overproduction of free radicals or from an insufficient production of antioxidant enzymes, leading to cellular damage and disease progression [98].

#### 3.3.3. *Theileria* spp.

In this subsection, only eight of the total articles reviewed focused on identifying Theileria spp. in its vertebrate host—cattle. The reported prevalence of the disease in this species ranged from 5.6% to 67%. Regarding the vector, only one study documented the presence of the parasite in ticks [30,38,41,46,49,51,56,58,59,93,99,100,101,102,103,104,105,106,107,108].

The most frequently reported *Theileria* species in Europe include *T. annulata*, *T. orientalis*, *T. buffeli*, *T. sergenti*, and *T. sinensis*. These pathogens are primarily transmitted by ticks such as *Rhipicephalus bursa*, *Rhipicephalus turanicus*, *Dermacentor marginatus*, and *Hyalomma lusitanicum* [30,38,41,43,46,49,51,58].

Due to the varying clinical manifestations caused by *Theileria* spp., researchers have divided the genus into two main groups. The first group includes species that induce host cell transformation: *T. parva*, *T. annulata*, *T. lestoquardi*, and *T. taurotragi*. The second group comprises non-transforming species, which do not alter host cells, such as *T. orientalis*, *T. mutans*, and *T. velifera*. This classification is based on the parasite’s ability to transform leukocytes, enabling infected cells to proliferate along with the parasite. Transforming species are associated with significant genetic modifications (e.g., expansion of gene families), while non-transforming species do not induce such cellular changes [109,110,111,112].

Within these two categories, host specificity and pathogen virulence play a crucial role in the genetic variability observed among *Theileria* species. Host immunity against *Theileria* spp. is largely driven by cellular immune responses, particularly those involving MHC class I molecules. Conversely, immune evasion by the parasite is facilitated by the high genetic diversity of its antigenic determinants, which can impair recognition by host T-cell receptors [109].

Notably, two of the reviewed studies reported the coexistence of multiple *Theileria* species in a single host. Bursakov and Kovalchuk (2019), using a combination of diagnostic methods, identified five *Theileria* species in cattle: *T. annulata*, *T. orientalis*, *T. buffeli*, *T. sergenti*, and *T. sinensis* [41]. In addition, co-infections involving Theileria spp. and other pathogens have been documented. For example, Zhou et al. (2016), using PCR, detected the simultaneous presence of *Anaplasma marginale*, *T. annulata*, *B. bigemina*, and *T. orientalis*, with respective prevalence rates of 29.1%, 18.9%, 11.2%, and 5.6% [38].

#### 3.3.4. *Borrelia* spp.

The genus *Borrelia* comprises approximately 50 species of Gram-negative spirochetes, grouped into four main phylogenetic clades: the relapsing fever group, reptile-associated *Borrelia*, monotreme-associated *Borrelia*, and the *Borrelia burgdorferi* sensu lato complex. Several of these species are considered highly pathogenic and have been identified in a wide range of mammalian hosts, particularly rodents and cervids [113,114,115]. In the temperate and colder regions of Europe—including Western, Central, Eastern, and Northern Europe—*Borrelia* spp. are recognized as important zoonotic pathogens transmitted by ticks [54].

In this review, 6 of the 36 analyzed articles specifically addressed borreliosis [32,36,40,42,44,54]. Two studies investigated the prevalence of the disease in cattle, reporting values ranging from 1.1% to 66%. The remaining four studies focused on the prevalence of *Borrelia* in tick populations, with infection rates between 13.8% and 27% [32,36,40,42,44,54].

The *Borrelia burgdorferi* sensu lato complex was mentioned in 12 studies—6 conducted in Germany and 1 each from France, Poland, Slovakia, Sweden, and Switzerland, as well as 1 study including data from Poland. These studies reported prevalence rates in cattle ranging from 1.1% in northern Sweden to as high as 66% in Germany [32].

The presence of *Borrelia burgdorferi* s.l., *Borrelia afzelii*, and *Borrelia garinii* in tick populations has been confirmed in both eastern and western Norway [36], as well as in Belgium [44]. Moreover, Adjadj and colleagues (2023) identified a case of coinfection involving Borrelia spp. and Anaplasma phagocytophilum in ticks [44].

Hvidsten et al. (2020) reported that the prevalence of *Borrelia* spp. in *Ixodes ricinus* ticks ranged between 1–15% and 15–27%, depending on the region [36]. Similarly, Sprong et al. (2020) confirmed the presence of two *Borrelia* species—*B. burgdorferi* s.l. and *B. miyamotoi*—in *I. ricinus* [42]. In the case of *Dermacentor reticulatus* ticks, *Borrelia afzelii* and *Anaplasma phagocytophilum* were found in association using PCR techniques [40].

In conclusion, the increasing prevalence of *Anaplasma* spp., *Babesia* spp., *Theileria* spp., and *Borrelia* spp. can be attributed to two main factors: the application of a broader range of diagnostic techniques, and the expansion of surveillance into areas where these pathogens had not been previously reported. This trend appears to correlate with the effects of global warming and the growing populations of vector species.

## 4. Discussion

The aim of this study was to provide an overview of the most significant zoonotic bacterial and parasitic diseases transmitted by ticks to domestic cattle in tropical and subtropical regions of Europe [54]. However, the number of studies dedicated to specific diseases in Europe remains relatively limited. Although the etiology of these diseases involves multiple bacterial and parasitic species, the studies reviewed primarily focused on only a few of these pathogens [54].

Based on the geographic regions covered in the analyzed literature, the following countries were included: Romania, Bosnia and Herzegovina, Croatia, Italy, Turkey, Portugal, Germany, France, Poland, Sweden, England, Ireland, Russia, the Netherlands, Belgium, and Spain.

The occurrence and spread of tick-borne diseases are directly dependent on the presence of competent vector species. Understanding the biology of these bacterial and parasitic pathogens requires the identification of one or more host species, and their geographic distribution may be narrower than that of their tick vectors [32]. Among the tick species involved in pathogen transmission, the most frequently reported—ranked in descending order—are *Ixodes ricinus*, *Hyalomma marginatum*, and *Dermacentor marginatus*.

An accurate diagnosis requires a combination of tests to adequately evaluate samples collected from animals. Advanced molecular diagnostic techniques, such as next-generation sequencing (NGS) and metagenomic analysis, offer promising improvements in the detection of novel and emerging pathogens, as they can identify multiple infectious agents in a single assay [116].

Another issue raised in several studies was the occurrence of single or multiple co-infections. According to research by Belongia (2002) and Diuk-Wasser et al. (2016), co-infections present significant diagnostic challenges, as pathogens may act synergistically or antagonistically within the host, influencing disease severity [117,118,119].

In conclusion, further research is urgently needed to identify these neglected pathogens within European countries. It is also essential to conduct expanded surveillance using highly sensitive and specific diagnostic methods, particularly in regions that have not been previously investigated or where research remains scarce. These studies are vital to determine whether and how global changes are influencing the frequency and distribution of neglected zoonotic pathogens transmitted by ticks to domestic animals [42].

## 5. Conclusions

The present study has highlighted the most frequently identified pathogenic species involved in the development of anaplasmosis, babesiosis, theileriosis, and borreliosis, as well as the key tick species responsible for the transmission of these for transmitting both zoonotic and non-zoonotic pathogens across Europe.

## Figures and Tables

**Figure 2 animals-15-01408-f002:**
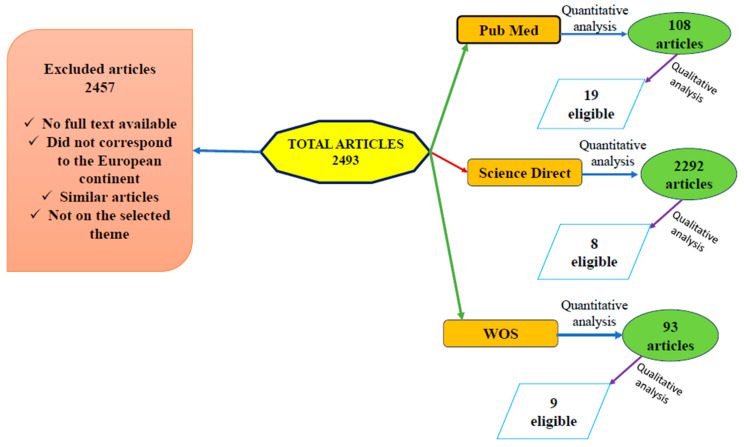
Diagram highlighting the selection technique applied to identify relevant items.

**Table 1 animals-15-01408-t001:** Diagnostic methods used to identify tick-borne pathogens in bovines, modified after [81].

Disease	Pathogen Name	Available Diagnostic Test	Geographical Area	Prevalence	References
Anaplasmosis	*A. phagocytophilum* *	Giemsa Staining Method, PCR	Great Britain	22.9%	[29,81]
	*A. phagocytophilum* **	PCR	Norway	1.0–13.5%	[32]
	*A. phagocytophilum* *	PCR	Sweden	23.9%	[33]
	*A. phagocytophilum* *	PCR	France	0.6%	[34]
	*A. phagocytophilum* **	PCR	France	1.75%	[34]
	*A. phagocytophilum* **	PCR	Germany	0.05%	[40]
	*A. bovis* *		Croatia		[30]
	*Anaplasma* spp. ***	ELISA	Belgium	15.6%	[44]
	*A. phagocytophilum* *	IFAT	Belgium	34.2%	[44]
	*A. phagocytophilum* **	RT-PCR	Belgium	0.5%	[44]
	*A. phagocytophilum* **	PCR	Spain	5.6%	[48]
	*A. centrale* **	PCR	Spain	1.4%	[48]
	*A. marginale* *	RT-PCR	Russia	42%	[55]
	*A. marginale* **	PCR	Portugal	2.6%	[58]
	*A. marginale* **	PCR	France	2%	[45]
	*A. marginale* *	PCR	France	6%	[45]
Babesiosis	*B. divergens* *	Giemsa Staining Method, PCR	Great Britain	57.1%	[29,81]
	*B. divergens* *	PCR	Bosnia and Herzegovina		[30]
	*B. divergens* *		Austria		[30]
	*B. divergens* *		Germany	13.0–21.0%	[30]
	*B. divergens* *	PCR	Sweden	53.5%	[33]
	*B. major* *		Germany		[30]
	*B. divergens* *		Hungary		[30]
	*Babesia spp* *	PCR	Poland	10.4%	[30]
	*B. major* **	PCR	Switzerland		[30]
	*B. occultans* *	RT-PCR	Poland	10.4%	[31]
	*B. bigemina* *	PCR	Turkey	11.2%	[38]
	*B. bovis* *	iELISA	Portugal	79%	[39]
	*B. bovis* *	PCR	Portugal	71%	[39]
	*B. bigemina* *	iELISA	Portugal	52%	[39]
	*B. bigemina* *	PCR	Portugal	34%	[39]
	*Babesia* spp. *	IFAT	Belgium	3.4%	[44]
	*Babesia* spp. **	PCR	England and Wales	0.38%	[47]
	*B. divergens* *	RT-PCR	Germany	6.5%	[52]
	*B. divergens* **	RT-PCR	Germany	0.9%	[52]
	*B. microti* **	PCR	Germany	0.49%	[54]
	*B. venatorum* **	PCR	Germany	0.42%	[54]
	*B. capreoli* **	PCR	Germany	0.07%	[54]
Theileriosis	*T. orientalis* *		Croatia		[30]
	*T. sergenti* *	PCR	Serbia	3.7%	[30]
	*Theileria* spp. ***	PCR	Serbia	3.7%	[30]
	*T. buffeli* *	PCR	Serbia	3.7%	[30]
	*T. orientalis* *		Hungary		[30]
	*T. orientalis* *	PCR	Turkey	5.6%	[38]
	*T. annulata* *	PCR	Turkey	18.9%	[38]
	*T. annulata* *	PCR	Spain	22.4%	[56]
	*T. annulata* **	PCR	Portugal	0.37%	[58]
	*T. annulata* *	PCR	Portugal	30%	[46]
	*T. orientalis* *	PCR	Hungary		[49]
	*T. annulata* *	PCR	Spain	22.4%	[56]
Borreliosis	*B. burgdorferi (s.l.)* **	PCR	Norway	0.17–24.2%	[32]
	*Borrelia* spp. ***	ELISA	Belgium	12.9%	[44]
	*B. burgdorferi sensu lato* **	RT-PCR	Belgium	13.8%	[44]
	*B. afzelii* **	RT-PCR	Belgium	65.7%	[44]
	*B. garinii* **	RT-PCR	Belgium	17.1%	[44]
	*B. burgdorferi* **	PCR	France		[45]
	*B. miyamotoi* **	PCR	Germany	0.25%	[40]
	*B. afzelii* **	PCR	Germany		[40]

* cattle, ** ticks.

**Table 2 animals-15-01408-t002:** Tick vectors implicated in pathogen transmission.

Parasitic/Bacterial Pathogen Agent	Vectors	References
*B. burgdorferi* sensu lato	*I. ricinus, H. marginatum*	[7,32,36,42,45,54]
*B. miyamotoi*	*I. persulcatus*, *D. reticulatus*	[40]
*Anaplasma* spp.	*R. turanicus*	[43]
*A. phagocytophilum*	*I. ricinus, H. punctata, R. pusillus, D. marginatus*, *H. marginatum,* and *Rhipicephalus* spp.	[7,9,32,34,42,44]
*A. bovis*	*H. punctata*	[44]
*A. marginale*	*R. bursa, R. turanicus, H. excavatum*	[58,82]
*B. divergens*, *B. venatorum*, *B. divergens/capreoli, B. odocoilei-like*	*I. ricinus*	[9,28,29,47]
*B. venatorum*	*I. ricinus*	[42]
*Babesia* spp.	*R. turanicus*/*D. marginatus, H. lusitanicum*	[50]
*B. bovis*	*R. turanicus, R. bursa, H. excavatum, H. anatolicum*	[82]
*B. occultans*	*H*. *excavatum*	[82]
*Theileria* spp.	*R. turanicus*/*D. marginatus, H. lusitanicum*	[43]
*Th. annulata*	*R. bursa*, *R. turanicus, H. excavatum*	[58,82]
*Th. sergenti/buffeli/orientalis*	*R. annulatus*	[50]

## Data Availability

Data are contained within the article.

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
