# Peer review of "Cattle Zoonotic and Non-Zoonotic Tick-Borne Pathogens in Europe—A Retrospective Analysis of the Past 15 Years"

_animals, 2025, doi:10.3390/ani15101408_

Round 1

Reviewer 1 Report

Comments and Suggestions for Authors

The results lack systematicity. Additional tables could be included in the manuscript to better organize and present the data.

In the "Diagnostic Methods" section, Polymerase Chain Reaction (PCR) and quantitative PCR (qPCR) are currently grouped together. However, this arrangement may not be appropriate, as they are distinct techniques with different applications. It would be more effective to separate them into subsections to clarify their roles and methodologies.

Additionally, it would be valuable to highlight which method is most commonly used for pathogen detection. This information would provide readers with a clearer understanding of current practices and trends in the field.

In the "Species of Ticks" section, it is crucial to specify which tick species act as vectors for the pathogens discussed. This information is vital for understanding the transmission dynamics and epidemiology of tick-borne diseases. Including a clear association between specific tick species and the pathogens they carry would significantly enhance the relevance and utility of this section.

In the "Tick-borne Pathogens" section, the paper reports that the prevalence of Anaplasma spp. in cattle populations ranges from 0.6% to 100%. However, to provide a more comprehensive understanding, it is essential to include details on the geographic distribution of the pathogen and its prevalence rates across different regions of the country. This information would help clarify the epidemiological patterns and highlight areas with higher infection risks, which is critical for targeted control and prevention strategies.

In summary, the results and discussion sections require careful revision to improve clarity, organization, and systematicity.

Comments on the Quality of English Language

English language should be improved.

Author Response

Dear Reviewer,

We would like to express our sincere gratitude for your invaluable time and consideration in reviewing this manuscript. Your meticulous attention to detail and constructive feedback have been instrumental in enhancing the quality of our work.

Comments 1:

The results lack systematicity. Additional tables could be included in the manuscript to better organize and present the data.

Response 1:

Line 683-769, 830-1035: Thank you for the recommendation. Please consult the updated text.

Comments 2:

In the "Diagnostic Methods" section, Polymerase Chain Reaction (PCR) and quantitative PCR (qPCR) are currently grouped together. However, this arrangement may not be appropriate, as they are distinct techniques with different applications. It would be more effective to separate them into subsections to clarify their roles and methodologies.

Response 2:

Line 503-589, 590-597: Thank you for the recommendation. We have allocated a paragraph for each method. Please consult the updated text.

Comments 3:

Additionally, it would be valuable to highlight which method is most commonly used for pathogen detection. This information would provide readers with a clearer understanding of current practices and trends in the field.

Response 3:

Line494-499: Thank you for the recommendation. We have formulated the following sentence and inserted it into the text: Across the 36 articles reviewed, several diagnostic methods were reported as being used to confirm the presence of the pathogens. Laboratory techniques, ranked in descending order of frequency, included: PCR (polymerase chain reaction), real-time PCR, clinical signs-based diagnosis, multiplex real-time PCR, nested PCR, serological tests such as ELISA, iELISA, and IFAT, as well as Giemsa-stained spleen smears and cloning.

Comments 4:

In the "Species of Ticks" section, it is crucial to specify which tick species act as vectors for the pathogens discussed. This information is vital for understanding the transmission dynamics and epidemiology of tick-borne diseases. Including a clear association between specific tick species and the pathogens they carry would significantly enhance the relevance and utility of this section.

Response 4:

Line 799-826: We would like to express our gratitude for your relevant comment. Please consult the updated text.

Comments 5:

In the "Tick-borne Pathogens" section, the paper reports that the prevalence of Anaplasma spp. in cattle populations ranges from 0.6% to 100%. However, to provide a more comprehensive understanding, it is essential to include details on the geographic distribution of the pathogen and its prevalence rates across different regions of the country. This information would help clarify the epidemiological patterns and highlight areas with higher infection risks, which is critical for targeted control and prevention strategies.

Response 5:

Line 1100-1147: Thank you for the recommendation. Please consult the updated text.

Comments 6:

In summary, the results and discussion sections require careful revision to improve clarity, organization, and systematicity.

Response 6: Thank you for the recommendation. Please consult the updated text

Reviewer 2 Report

Comments and Suggestions for Authors

The current review article "Cattle zoonotic tick-borne diseases in Europe - a retrospective 2 analysis of the past 15 years" is an important piece of work, culminating and identifying the tick-borne disease load in cattle research done over 15 years. The introduction is clearly written and clarifies the need of the Review in the field. I have minor concerns as follows:

  1. Please write a proper legend for Fig. 1 describing the color in the map and the geographical location magnified in the map
  2. The different diagnostic tools by the 36 papers chosen after filtering have different levels of sensitivity,. The authors must comment on the sensitivity of these diagnostic tools and thus, acknowledge that this limits the 100% accuracy of data presented here.
  3. The authors can re-examine the title: these are zoonotic infections but the phrasing "cattle zoonotic tick-borne disease...." does not seem appropriate. 

Author Response

Dear Reviewer,

We would like to express our sincere gratitude for your invaluable time and consideration in reviewing this manuscript. Your meticulous attention to detail and constructive feedback have been instrumental in enhancing the quality of our work.

Comments 1:

Please write a proper legend for Fig. 1 describing the color in the map and the geographical location magnified in the map.

Response 1:

Line 475-477: Thank you for the recommendation. Please consult the text below Figure 1, which has been updated.

Comments 2:

The different diagnostic tools by the 36 papers chosen after filtering have different levels of sensitivity,. The authors must comment on the sensitivity of these diagnostic tools and thus, acknowledge that this limits the 100% accuracy of data presented here.

Response 2:

Line 585-589, 594-597, 604-608, 614-621, 624-655, 661-664, 671-678: Thank you for the recommendation. Please consult the updated text.

Comments 3:

The authors can re-examine the title: these are zoonotic infections but the phrasing "cattle zoonotic tick-borne disease...." does not seem appropriate.

Response 3:

Line 2-3: Thank you for the recommendation. Please consult the updated title.

Reviewer 3 Report

Comments and Suggestions for Authors

The authors performed a systematic, retrospective review of cattle tick-borne diseases in Europa in the last 15 years. This is a timely and relevant study, given the negative economic  impact that tick-borne diseases can pose to the cattle industry in Europe and globally. The authors emphasize some aspects associated with the zoonotic importance of the tick-borne transmitted pathogens, and this is of particular importance, considering the concept of One-Health. The Abstract, Introduction, and M&M are well written. However, the Results and Discussion sections need major modifications. My comments and suggestions are listed below.

  • In the Intro, add references to lines 63-75.
  • In the Intro, add references to lines 92-108.
  • Move paragraph of lines 54-62 to the end of the Introduction section.
  • The authors report an increase in the prevalence of Anaplasma, Babesia, Theileria, etc., however, it is not clear if this increase is real or caused by the more availability of diagnostics. The authors need to discuss this issue.
  • The concept of One-Health is important, and it is highlighted in the title of the manuscript. However, very little discussion is presented in the manuscript, regarding the pathogens that have potential zoonotic risks. The authors need to emphasize the importance of B. divergens as a causative agent of human babesiosis. Additionally, the authors must clarify that so far, no Theileria species have been demonstrated to infect and cause diseases in humans.
  • In the Results section, lines 167 to 202 should be deleted. These are explanations about molecular/serological assays and do not fit in the scope of the manuscript.
  • The sub-sections of the Results need a “take-home message.” This will help readers to understand the most important message of the data.
  • Results, sub-sections 3.3.3 and 3.3.4: These sections need to be reformatted; as they’re now, these are disconnected sentences. 
  • The Discuss section is not reading well. Sentences are disconnected with short paragraphs. It needs a flow to be attractive for readers.

Author Response

Dear Reviewer,

We would like to express our sincere gratitude for your invaluable time and consideration in reviewing this manuscript. Your meticulous attention to detail and constructive feedback have been instrumental in enhancing the quality of our work.

Comments 1:

In the Intro, add references to lines 63-75

Response 1:

Line 124-136: Thank you for the recommendation. Please consult the updated text.

Comments 2:

In the Intro, add references to lines 92-108.

Response 2:

Line 200-213: Thank you for the recommendation. Please consult the updated text.

Comments 3:

Move paragraph of lines 54-62 to the end of the Introduction section.

Response 3:

Line 442-450: Thank you for the recommendation. Please consult the updated text.

Comments 4:

The authors report an increase in the prevalence of Anaplasma, Babesia, Theileria, etc., however, it is not clear if this increase is real or caused by the more availability of diagnostics. The authors need to discuss this issue.

Response 4:

Line 1390-1394: Thank you for the recommendation. Please consult the updated text.

Comments 5:

The concept of One-Health is important, and it is highlighted in the title of the manuscript. However, very little discussion is presented in the manuscript, regarding the pathogens that have potential zoonotic risks. The authors need to emphasize the importance of B. divergens as a causative agent of human babesiosis. Additionally, the authors must clarify that so far, no Theileria species have been demonstrated to infect and cause diseases in humans.

Response 5:

Line 104-114, 229-249, 1039-1041 : Thank you for the recommendation. Please consult the updated text.

Comments 6:

In the Results section, lines 167 to 202 should be deleted. These are explanations about molecular/serological assays and do not fit in the scope of the manuscript.

Response 6:

Line 493-502: Thank you for the recommendation. Please consult the updated text.

Comments 7:

The sub-sections of the Results need a “take-home message.” This will help readers to understand the most important message of the data.

Response 7: Thank you for the recommendation. Â

Line 679-681, 1041-1043, 1390-1394

Comments 8:

Results, sub-sections 3.3.3 and 3.3.4: These sections need to be reformatted; as they’re now, these are disconnected sentences.

Response 8: Thank you for the recommendation. Please consult the updated text.

Line 1298-1359: 3.3.3. Theileria spp.

Line 1361-1394: 3.3.4. Borrelia spp.

Comments 9:

The Discuss section is not reading well. Sentences are disconnected with short paragraphs. It needs a flow to be attractive for readers.

Response 9: Thank you for the recommendation. Please consult the updated text.

Line 1715-1747

Round 2

Reviewer 1 Report

Comments and Suggestions for Authors

The paper has been improved.

Author Response

Dear Reviewer, 

Thank you for the favorable review.